# Tetrodotoxin Decreases the Contractility of Mesenteric Arteries, Revealing the Contribution of Voltage-Gated Na^+^ Channels in Vascular Tone Regulation

**DOI:** 10.3390/md21030196

**Published:** 2023-03-22

**Authors:** Joohee Park, Coralyne Proux, William Ehanno, Léa Réthoré, Emilie Vessières, Jennifer Bourreau, Julie Favre, Gilles Kauffenstein, César Mattei, Hélène Tricoire-Leignel, Daniel Henrion, Claire Legendre, Christian Legros

**Affiliations:** 1INSERM, CNRS, MITOVASC, Equipe CarME, SFR ICAT, University Angers, 49000 Angers, France; 2UMR INSERM 1121, CRBS, Strasbourg University, 67000 Strasbourg, France; 3UMR INSERM 1260, CRBS, Strasbourg University, 67084 Strasbourg, France

**Keywords:** TTX, voltage-gated Na^+^ channels, aorta, mesenteric arteries, veratridine, RT-qPCR

## Abstract

Tetrodotoxin (TTX) poisoning through the consumption of contaminated fish leads to lethal symptoms, including severe hypotension. This TTX-induced hypotension is likely due to the downfall of peripheral arterial resistance through direct or indirect effects on adrenergic signaling. TTX is a high-affinity blocker of voltage-gated Na^+^ (Na_V_) channels. In arteries, Na_V_ channels are expressed in sympathetic nerve endings, both in the intima and media. In this present work, we aimed to decipher the role of Na_V_ channels in vascular tone using TTX. We first characterized the expression of Na_V_ channels in the aorta, a model of conduction arteries, and in mesenteric arteries (MA), a model of resistance arteries, in C57Bl/6J mice, by Western blot, immunochemistry, and absolute RT-qPCR. Our data showed that these channels are expressed in both endothelium and media of aorta and MA, in which *scn2a* and *scn1b* were the most abundant transcripts, suggesting that murine vascular Na_V_ channels consist of Na_V_1.2 channel subtype with Na_V_β1 auxiliary subunit. Using myography, we showed that TTX (1 µM) induced complete vasorelaxation in MA in the presence of veratridine and cocktails of antagonists (prazosin and atropine with or without suramin) that suppressed the effects of neurotransmitter release. In addition, TTX (1 µM) strongly potentiated the flow-mediated dilation response of isolated MA. Altogether, our data showed that TTX blocks Na_V_ channels in resistance arteries and consecutively decreases vascular tone. This could explain the drop in total peripheral resistance observed during mammal tetrodotoxications.

## 1. Introduction

Voltage-gated Na^+^ (Na_V_) channels are members of the voltage-gated ion channel superfamily, consisting of pore-forming α-subunits (Na_V_1.1-9) encoded by nine distinct genes (*scn1a*-*5a*, *scn8*-*11a*) with one or two auxiliary subunits (Na_V_β1-4) encoded by four genes (*scn1b*-*4b*) in mammals [1]. By controlling the Na^+^ currents through the plasma membrane of excitable and non-excitable cells, Na_V_ channels exhibit different physiological roles, such as electrical coupling, muscle and heart contraction, or hormone secretion [1,2].

In the vasculature, Na_V_ channels are expressed and generate Na^+^ currents in both endothelial cells (EC) and vascular smooth muscle cells (VSMC) isolated from the aorta, pulmonary, mesenteric, coronary, uterine, and cremaster arteries [3,4,5,6,7,8,9,10,11,12,13,14,15,16]. Their involvement in vascular contractility has been demonstrated in rat mesenteric arteries (MA) [9]. Moreover, Na_V_ channels endogenously expressed in EC have been suggested to participate in conducted vasodilator response in murine cremaster arteries [13]. In addition, in vitro studies have shown that endothelial Na_V_ channels are activated by shear stress [16] and participate in VEGF-MAP-kinase signaling associated with proliferation, migration, and tubulogenesis of EC [14]. Finally, vascular Na_V_ channels interfere with various features of vascular function, through a functional coupling with Na^+^/Ca^2+^ exchanger (NCX), mediating Ca^2+^ entry in both VSMC and EC [9,14].

Na_V_ channels are pharmacologically classified according to their sensitivity to tetrodotoxin (TTX): TTX-sensitive (TTX-S) Na_V_ channels (IC_50_ in the nanomolar range for Na_V_1.1-4 and Na_V_1.6-7) and TTX-resistant (TTX-R) Na_V_ channels (IC_50_ in the micromolar range for Na_V_1.5 and Na_V_1.8-9) [17]. TTX poisoning, also called tetrodotoxication, is an important health issue due to fish consumption and particularly in puffer fish eaters [18]. Deep hypotension is one of the most severe symptoms of human poisonings, which has been attributed to the reduction of vascular tone [18,19]. In rats, TTX-induced hypotension can be overcome by the α1-adrenergic agonist, phenylephrine (Phe), suggesting indirect effects of TTX on vascular tone [20]. These indirect effects are likely due to the inhibition of noradrenaline (NA) release by arterial sympathetic neurons. To date, the effects of TTX have not been investigated on isolated arteries, and one can not exclude that TTX might also act on arterial Na_V_ channels.

In this work, we characterized the gene expression profile of α and β Na_V_ channel subunits in a conductance artery (aorta) and a model of resistance arteries (MA) from mice. By using Western blot, immunochemistry, and absolute RT-qPCR, we showed that both arteries mainly expressed the Na_V_1.2 channel subtype with Na_V_β1. We then challenged isolated MA by myography and arteriography with TTX to investigate the role of Na_V_ channels in vascular tone to address their putative role in tetrodotoxication.

## 2. Results

### 2.1. Nav Channels Are Expressed in Both Intima and Media of Murine Arteries

In this work, we used two distinct branches of MA, which constitute the vascular bed of the small intestine, called first-order MA (FOMA) and cecocolic arteries (CA) (Figure 1a). The expression of Na_V_ channels in MA was evidenced by Western blot in FOMA, CA, and aorta using an anti-PanNa_V_ antibody that strongly immunostained 2 proteins of ~250 and ~280 kDa (Figure 1b). The 280 kDa band corresponds to that observed with brain protein extracts, which mainly contain the Na_V_1.2 subtype [1]. The expression of Na_V_ channels in MA was validated by immunohistochemistry in FOMA and CA (Figure 1c,d). This immunohistochemical analysis shows a strong immunofluorescence labeling with an anti-PanNa_V_ antibody in both intima and media of FOMA and CA (Figure 1c,d). The localization of Na_V_ channels in the endothelium was confirmed using anti-PECAM1 antibodies (Figure 1c,d; merged image). Beyond PECAM1 staining, the immunofluorescence labeling with anti-PanNa_V_ antibody likely corresponds to VSMC that expresses Na_V_ channels.

The expression of Na_V_ channels was also characterized in the aorta of C57Bl/6J mice (Figure 2a). Na_V_ channels were clearly immunodetected in the aorta using an anti-PanNa_V_ antibody (Figure 2b). In addition, the Na_V_β1 subunit was immunolabeled in the aorta (Figure 2c). Altogether, our immunohistochemical data indicate that both Na_V_ channel α- and β-subunits are expressed at the protein level in the intima and media of both MA and aorta from mice.

To identify which Na_V_ channel isoforms are expressed in the aorta, FOMA, and CA in mice, we used RT-qPCR with absolute quantification. The data are shown in Table 1 and Figure 3. We amplified *nos3* and *desmin*, encoding the endothelial NO synthase (eNOS) and desmin, respectively, to show that our RNA extraction is a mixture of RNA from both media and intima (Figure 3a). As expected, the expression levels of *desmin* were high and particularly in FOMA and CA, reflecting a higher content of desmin in resistance arteries, which contain more VSMC (Figure 3a). In comparison to *desmin*, the expression levels of *nos3* were found at much lower levels, certainly, because intima is organized in a single cell layer. In the aorta, FOMA, and CA, the following genes, encoding Na_V_ α-subunit were not detected: *scn1a*, *scn4a*, *scn8a*, *scn9a*, *scn10a*, and *scn11a*. The *scn2a* transcript appeared to be the most abundant and particularly in the aorta, compared to FOMA and CA (Figure 3b, Table 1). The *scn3a* transcript was observed only in the aorta, while *scn5a* was found only in FOMA and CA but at low levels for both genes (Figure 3b). The *scn1b* transcript was revealed with the highest level of expression compared to the other β-subunit genes (Figure 3b, Table 1). The *scn2b* transcript is expressed at similar levels in the aorta, FOMA, and CA. Low expression levels of *scn3b* and *scn4b* transcripts were detected. Altogether, these RT-qPCR data showed that *scn2a* and *scn1b* transcripts are the most abundant Na_V_ channel subunit genes expressed in the murine aorta, FOMA, and CA. This suggests that these arteries express the TTX-S Na_V_1.2 channels associated with Na_V_β1.

### 2.2. Effects of TTX on Murine Mesenteric Artery Contractility

To explore the implication of Na_V_ channels in the vascular tone of MA, we first investigated the effects of TTX on the contractile response of CA, a branch of MA by myography (Figure 4). We used veratridine (VTD) to activate all Na_V_ channels in arteries, including those expressed in terminal nerve endings. To avoid any effects of neurotransmitter release, such as NA, acetylcholine (Ach), and ATP, we used prazosin (PZ), atropine (AP), and suramin (SUR) to antagonize α1-adrenoreceptors, muscarinic, and ATP receptors, respectively, in order to discriminate the effects of VTD on vascular Na_V_ channels and then challenged the action of TTX (Figure 4a). Without VTD, TTX has no effect on CA contractility (data not shown). The application of VTD did not significantly modify the contractile response induced by U46619, a thromboxane A2 analogue (a potent vasoconstrictor), in the following conditions: (1) in the presence of PZ and AP (*p* > 0.99, Figure 4b) and (2) in the presence of PZ, AP, and SUR (*p* = 0.34, Figure 4c). However, the addition of TTX triggered a fast and almost complete relaxation of arteries to a similar extent compared to control: 88.4 ± 6.5% (*p* < 0.05) and 92.5 ± 1.6% (*p* < 0.01), respectively (Figure 4b,c). Thus, there was no significant difference in TTX-induced relaxation in the absence or the presence of SUR (*p* = 0.94, Mann–Whitney test), indicating that VTD stimulation did not trigger the release of a sufficient amount of ATP to modify the vascular tone. In both cases, after washout, CA gave similar responses to U46619 and TTX. In conclusion, these data show that TTX induced a full relaxation of CA, suggesting the activation of vascular Na_V_ channels by VTD.

### 2.3. Effects of TTX on Dilation Capacity of Murine Mesenteric Arteries

In order to investigate the effects of TTX on endothelial-mediated dilation, we performed arteriography experiments on isolated FOMA and CA (Figure 5). For each artery, the initial contraction level induced by Phe was not significantly different in the presence or the absence of TTX (Figure 5a,b; left panels). As expected, CA and FOMA progressively developed flow-mediated dilation (FMD) from 6 µL/min to 50 µL/min (Figure 5a,b; right panels). Interestingly, we observed that TTX significantly potentiated FMD responses of both CA and FOMA. TTX induced a significant potentiation (*p* < 0.05) of the dilation capacity from 6 to 30 µL/min and 6 to 50 µL/min for CA and FOMA, respectively (Figure 5a,b; right panels). For CA, the highest potentiation of TTX-induced dilation was observed at 6 µL/min compared to the control (2.9-fold, *p* < 0.05). Concerning FOMA, TTX potentiated dilation up to 2.7-fold (Figure 5a,b; right panels). In conclusion, TTX potentiates FMD of CA and FOMA, revealing the contribution of vascular Na_V_ channels to vascular tone regulation in response to flow increase.

## 3. Discussion

In this study, we evidenced that Na_V_ channels are expressed at the protein level in both intima and media of arteries from C57Bl/6J mice, including the aorta and MA. We determined that *scn2a* and *scn1b* encoding Na_V_1.2 and Na_V_β1 subunits are the major isoforms expressed in both the aorta and MA. To unmask the contribution of vascular Na_V_ channels to vascular tone, we pretreated isolated MA with VTD in the presence of specific antagonists to prevent the effects of neurotransmitters released in response to neuronal Na_V_ channel activation. In these conditions, TTX induced a total relaxation of MA. More interestingly, we found that TTX potentiates the dilation response to increasing flows. Thus, our data bring novel evidence that vascular Na_V_ channels contribute to the vascular tone and FMD, which could explain why TTX induces severe hypotension after tetrodotoxication.

Our data showed that Na_V_ channels are expressed in both endothelium and media of murine FOMA and CA in agreement with previous investigations performed on murine cremaster arteries [13], human subcutaneous arterioles [21], and human uterine arteries [15]. Our RT-qPCR data indicate that *scn2a*, encoding Na_V_1.2, is the most abundant Na_V_ channel α-subunit gene expressed in both aorta and MA. This observation agrees with the immunodetection of a protein with similar molecular weight detected in the brain by Western blot and other reports. The Na_V_ channel gene, *scn2a* is also well expressed in mouse cremaster [13] and femoral arteries, the MA and aorta from rats [6,9]. However, in humans, *scn9a* encoding Na_V_1.7 is the vascular Na_V_ channel isoform expressed in both uterine arteries and subcutaneous arterioles [15,21], but this subtype has not been detected in our mouse arteries. The *scn3a* transcript, encoding Na_V_1.3, is present in the aorta, but at a low level, suggesting that this subtype could be weakly expressed at the protein level. The same conclusion can be made for the *scn5a* transcript, encoding the cardiac Na_V_1.5, detected only in MA. Further experiments would be necessary to confirm the expression of these two Na_V_ channel subtypes. Indeed, low transcript expression level does not always reflect that of protein. However, the validation of Western blot or immunolocalization of a particular Na_V_ channel subtype with specific antibodies requires the use of negative controls, such as a knock-out mouse. The 250 kDa protein immunodetected in the aorta and MA could be a different glycosylated form of the Na_V_1.2 channel or a shorter spliced variant (isoform X2, GenBank: XP_006498646.1).

We also showed that *scn1b* encoding the Na_V_β1 subunit is the most abundant β-subunit expressed in both murine aorta and MA. In addition, *scn2b* and *scn3b* are also expressed but to a much lower extent. Indeed, the expression of Na_V_β subunits has hardly ever been studied in arteries, but similar to our data, *scn1b*, *scn2b*, and *scn3b* expressions have been reported in rat aorta [6], while *scn1-4b* are expressed in human uterine arteries [15]. Altogether, it seems that the expression profile of the Na_V_ channel gene in arteries is different in rodents and humans. Rodent arteries predominantly express *scn2a*, encoding Na_V_1.2, while human arteries mainly express *scn3a* and *scn9a*, encoding Na_V_1.3 and Na_V_1.7 subtypes, respectively.

We brought evidence that TTX caused full arterial relaxation, unmasking the contribution of arterial Na_V_ channels to vascular tone regulation. This effect was induced at a relatively low concentration of TTX (1 µM), suggesting that TTX-induced vasorelaxation is mainly mediated by TTX-S Na_V_ channels. So far, the contribution of Na_V_ channels to contraction has been explored by a pharmacological approach using VTD on isolated arteries [6,9,15,22]. However, VTD can activate all Na_V_ channels, including those of perivascular nerve endings, which leads to the release of neurotransmitters, such as NA, Ach, ATP, or neuropeptide Y [23], etc. In this way, the inhibition of α1-adrenoreceptor by PZ abolishes the VTD-induced contractile response of rat aortic rings [6] and of human uterine arteries [15], while in the presence of this antagonist, a weaker VTD-induced contractile response is still seen in MA from rat [9]. In our study, we showed for the first time that TTX can induce a vasorelaxation of arteries, revealing a putative contribution of Na_V_ channels expressed in VSMC and/or EC to vascular tone regulation.

The mechanosensitivity of Na_V_ channels is supported by strong evidence that membrane deformations can trigger Na_V_ channel activation [24,25,26]. Moreover, this response is regulated by both Na_V_β1 and Navβ3 subunits [27]. Interestingly, we found that TTX induces a potent relaxation of pressurized murine MA, revealing the contribution of Na_V_ channels to arterial mechanotransduction response to shear stress. Hence, we hypothesize that Nav channels are activated in VSMC in pressurized arteries and likely contribute to developing myogenic tone by mechanotransduction in physiological conditions [28]. Further experiments on isolated VSMC would help to show whether the electrophysiological properties of Na_V_ channels could be modulated by mechanical stimulation. Then, if Na_V_ channel stimulation induces a Ca^2+^ increase in VSMC and subsequent increase of myogenic tone, its blockade by TTX would lead to vasorelaxation. The contribution of Na_V_ channels to the crosstalk between Na^+^ and Ca^2+^ homeostasis has been clearly demonstrated in human coronary myocytes in which VTD triggers Ca^2+^ increase [11].

On the other hand, the vasorelaxant effects of TTX could also be mediated by its binding to endothelial Na_V_ channels and, thus, by inducing vasodilator factor release. Our data from arteriography clearly suggest that they contribute to the shear stress response. Indeed, it has been shown that Na_V_ channels expressed in HUVEC are activated by shear stress, which in turn could lead to a Ca^2+^ increase through NCX [16]. However, subsequent intracellular Ca^2+^ increase by Na_V_ channel activation has not been proven in EC. In HUVEC stimulated by VEGF, it has been shown that TTX induces an intracellular Ca^2+^ increase [14]. Since the increase of intracellular Ca^2+^ is an important pathway of NO synthase activation [29], this mechanism could be involved in the vasorelaxant effects of TTX observed in MA in response to flow increase. In addition, the activation of Na_V_ channels by shear stress limits Erk1/2 phosphorylation [14]. Since Erk1/2 could favor eNOS activation, this could explain why TTX increases vasodilation. However, the regulation of eNOS by Erk1/2 is controversial [30,31]. Further experiments are required to evaluate this hypothesis with isolated EC.

More recently, it has been shown that TTX suppresses a spontaneous contractile oscillation (vasomotion) of human uterine arteries under hypoxia, showing the contribution of Na_V_ channels to vasomotion through an electrogenic action in VSMC [15]. This assumption is reinforced by other data showing that TTX strongly reduces the upstroke of action potential induced by electrical stimulation in arteriolar SMC from murine skeletal muscle [32]. Hence, VSMC Na_V_ channels might be electrically activable and likely participate in the membrane depolarization leading to sarcolemmal Ca^2+^ entry and subsequent arterial myocyte contraction.

The effect of TTX on isolated arteries could partially explain why the arterial pressure drops down during tetrodotoxication [19]. Of course, TTX could reduce the sympathetic tone and, thus, the cardiac and vascular stimulation by decreasing NA release by sympathetic nerves and also adrenaline released by the adrenal medulla [33,34]. However, our data suggest that since TTX decreases contractility in isolated arteries, this toxin might also cause a significant reduction of total peripheral resistance and subsequent arterial hypotension, as observed during tetrodotoxication in dogs [19].

As suggested in the literature, we assume that in physiological conditions, Na_V_ channels are inactive or silent and do not contribute to physiological vascular responses [6,9,15]. In EC, patch clamp data indicate that Na_V_ channels are likely unavailable at resting potential [4,14,35], thus hyperpolarization is required to make them recover from inactivation. This could occur in tetrodotoxication, leading to hypoxia, which is known to activate K_ATP_ channels [15,36]. In VSMC, the situation is different since it has been shown that Na_V_ channel window currents are measurable at resting potential, suggesting their implication in action potential [5,10,11,12,32]. However, in isolated arteries, action potentials are insensitive to TTX in normal conditions [32]. Altogether, these data highlight that vascular Na_V_ channels are silent in normal conditions. Thus, since respiratory issues (dyspnea, hypoxia) occur during tetrodotoxication, this could favor the activation of these silent vascular Na_V_ channels.

## 4. Materials and Methods

### 4.1. Chemicals and Reagents

TTX was from Latoxan (Valence, France). VTD was from Santa Cruz Biotechnology (Dallas, TX, USA). Sodium nitroprusside and Optical Cutting Temperature (OCT) compound were from Thermo Fisher Scientific (Waltham, MA, USA). All other reagents, such as papaverine, Ach, Phe, PZ, AP, and SUR, were obtained from Sigma-Aldrich Merck (Saint-Louis, MO, USA). U46619 was from Cayman chemical (Ann Arbor, MI, USA).

### 4.2. Ethical Approval, Animals and Artery Preparations

All animal experiments were conducted in accordance with the European Community council directive 2010/63/EU for the care and use of laboratory animals. The experimental procedures used in this work were approved by our respective local ethical committees in addition to the French Ministry of Agriculture (N° A49007002, APAFIS#24014-202001311653562 authorization). The NC3R’s ARRIVE guidelines were followed in the conducting and reporting of all experiments using animals. Five-month-old male and female mice C57Bl/6J (Janvier Labs, France) were used in this work. Animals were housed in individual cages (internal dimensions in mm: 369 × 165 × 132, L × W × H for 5 mice) in controlled temperature rooms (21–23 °C) of the Animal Facilities Unit of the Hospital University of Angers (SCAHU). They were maintained under a 12 h light/dark cycle with free access to water and food.

Animals were euthanized in a CO_2_ chamber (Tem Sega, Pessac, France) with a concentration gradient: a first phase was carried out by mixing air with CO_2_ for 1 min with a flow rate of 10 L/min, and death was induced by pure CO_2_ for 1 min with a flow rate of 10 L/min. Then, aorta and MA were quickly harvested and placed in ice-cold physiological salt solution (PSS, in mM: NaCl, 130; KCl, 3.7; MgSO_4_(7H_2_O), 1.2; NaHCO_3_, 14.9; CaCl_2_(2H_2_O), 1.6; HEPES, 5; KH_2_PO_4_, 1.2; and D-glucose, 11). The pH of PSS was adjusted at 7.4 (pO_2_ 160 mmHg and pCO_2_ 37 mmHg). Adherent tissues of arteries were manually removed by the same experimenter. For immunohistochemical characterization, several segments of MA (Figure 1a) and aorta (Figure 2a) were conserved at −80 °C in embedding medium (OCT compound). To characterize the Na_V_ channel expression by RT-qPCR, arteries were stored at −20 °C in Allprotect Tissue Reagent (Qiagen, Germany). For myography and arteriography, freshly harvested MA was used.

### 4.3. Western Blot

CA and FOMA were isolated from 5-month-old C57Bl/6J males. To extract total proteins, arterial tissus were ground in liquid nitrogen and then lysed in 50 µL of 1% SDS extraction buffer (1% SDS, 10 mM Tris-base, 5 mM EDTA and protease inhibitor cocktail (Thermo Fisher Scientific)). After homogenization for 30 min at 8 °C, the sample was centrifuged at 14,000× *g* at 10 °C for 20 min. Supernatant protein quantity was determined using the Micro BCA Protein Assay Kit (Thermo Fisher Scientific). A total of 10 µg of extracted proteins were separated by SDS-PAGE (8% polyacrylamide gel) and transferred to a nitrocellulose membrane. Tris-buffered saline solution (0.1% Tween and 5% BSA) was used for 90 min to block the non-specific binding sites. Next, the membranes were incubated with the rabbit anti-PanNa_V_ primary antibody (1:500, ASC-003, Alomone Labs) overnight at 4 °C. The anti-HSC70 primary antibodies (1:10,000, SC-7298, Santa Cruz Biotechnology) were used as a loading control. After washing, the membranes were incubated with the anti-rabbit secondary antibodies conjugated to horse-radish peroxidase (1:5000, Thermo Fisher Scientific) for 90 min. Proteins were revealed using Pierce ECL Western Blotting Substrate kit (Thermo Fisher Scientific) by LAS-3000 imager (Fuji, Tokyo, Japan). The image acquisition was performed with Image Labs (Bio-Rad, Hercules, CA, USA).

### 4.4. Immunohistochemistry

Aorta and MA (from males) frozen in embedding medium were transversally cryosectioned at 10 or 12 µm thick by Cryostat (Leica) at −25 °C. Artery slices were transferred to SuperFrostPlus slides (Thermo Fisher Scientific, USA) and dried at room temperature. Immunofluorescence staining is processed using the rabbit primary anti-PanNa_V_ (1:200, ASC-003, Alomone labs) or the rabbit primary anti-Na_V_β1 (1:100, ASC-041, Alomone labs), and/or the rabbit primary anti-PECAM1 antibodies (anti-CD31 labeled with FITC, 1:50, eBioscience, San Diego, CA, USA) overnight at 4 °C. Nuclei were stained with DAPI 10 µg/mL (Molecular probes, Invitrogen). After washing, arteries were incubated with Alexa fluor 568 conjugated goat anti-rabbit antibodies (1:200, Invitrogen) for 3 h. After washing, the aqueous mounting medium (Fluoromount, Sigma-Aldrich Merck, Saint-Quentin-Fallavier, France) was used before placing a lamella on slide. Fluorescent staining was visualized using a confocal microscope Nikon Eclipse TE2000-S. The imaging acquisition was performed with Metamorph software (Molecular Devices, San Jose, CA, USA). Negative control experiments excluding the primary antibody were performed to verify the specificity of the immunofluorescence labeling.

### 4.5. RNA Extraction and RT-qPCR

Total RNAs were extracted from arteries using the RNeasy micro kit (Qiagen) as described, previously [37]. At least 300 ng (up to 1 µg, depending on RNA extraction yield) of total RNA were used for cDNA synthesis using random hexamers and the QuantiTect Reverse Transcription kit (Qiagen). RT-qPCR experiments were run on a LightCycler 480 Instrument II (Roche, Meylan, France) using Sybr^®^ Select Master Mix (Applied Biosystems^®^). PCR was performed with 6 ng of cDNA in duplicate and Na_V_ channel gene-specific primers (Table 2), which were designed using the Primer3 Software (https://github.com/primer3-org, accessed on 21 February 2023). Differences in transcript expression levels were determined using the cycle threshold method, as described earlier [38]. Peak-melting curve was performed at the end of the amplification process to check amplification specificity, and amplicon sizes were verified by agarose gel electrophoresis. For absolute quantification method, synthetic cDNA spanning each PCR amplicons were cloned into pUC57 or its derivatives. All cDNA sequences were verified by sequencing (DNA sequencing facility of Angers CHU) and compared to those available in the GenBank. For absolute quantification, cDNA copies were evaluated using the calibration curve method, which consists in the recombinant double-stranded plasmid DNA molecule determination as already described [39]. The cDNA copy numbers of 8 dilutions of pure plasmids were used to build the calibration curves for each gene, allowing PCR efficiency determination (~100% for each gene). Data from male (*n* = 5–6) and female (*n* = 6) were grouped.

### 4.6. Myography

CA was macrodissected under microscope from MA of 5-month-old male mice (Figure 1a). Arterial rings of 2 mm length were mounted in a Mulvany–Halpern-type wire myograph (Danish MyoTechnology, DMT, Aarhus, Denmark). Then, 2 tungsten wires with 40 µm diameter were inserted into the arterial lumen and fixed to a micrometer and a force transducer. Arterial segments were bathed in a 5 mL PSS maintained at 37 °C and pH of 7.4 (pO_2_ 160 mmHg and pCO_2_ 37 mmHg). Arterial wall tension was normalized, and then, arteries were stabilized for 40 min. To evaluate the maximal contractile capacity, a K^+^ rich PSS (80 mM) with 3 µM Phe was used. To control the endothelial integrity, contraction was induced with 2 µM Phe, and then, 2 µM Ach was added to elicit vasorelaxation. Two cocktails of antagonists were then added in order to prevent any effects of neurotransmitter release (NA, Ach, and ATP) by terminal nerve endings: 1 µM PZ with 100 nM AP (PZ + AP) or 1 µM PZ with 100 nM AP with 100 µM SUR (PZ + AP + SUR). Afterwards, U46619 (10 to 80 nM, thromboxane A2 analogue) was applied to reach 50 to 70% of maximal contraction, followed by VTD at 30 µM. After the stabilization of VTD-induced effects, TTX was added to the bath. Since both TTX-S and TTX-R Na_V_ channel gene transcripts have been detected by RT-qPCR in MA, we chose 1 µM TTX in order to affect both kinds of Na_V_ channels. At this concentration, we expected partial effects if TTX-R Na_V_ channels would contribute to arterial response or a full response mediated by TTX-S Na_V_ channels.

### 4.7. Arteriography

MA was harvested from 5-month-old male and female mice. CA and FOMA (Figure 1a) were canulated at both ends on glass microcannulas mounted in a video-monitored perfusion system [40] (Living System, LSI, Burlington, VT, USA). Each artery segment was bathed in a 10 mL PSS maintained at 37 °C and at pH of 7.4 (pO_2_ 160 mmHg and pCO_2_ 37 mmHg). The perfusion of the arteries was monitored with two peristaltic pumps, one controlling the flow rate and the other under the control of a pressure-servo control system. The pressure was set at 75 mmHg. Contraction to Phe (1 µM) and dilation to Ach (1 µM) were assessed. Afterwards, to evaluate FMD, we contracted the arteries with Phe (1 µM) under 3 µL/min and then, increased flow (6 to 50 µL/min) by steps. To investigate the involvement of Na_V_ channels in FMD, the experiments were carried out in the absence or the presence of TTX (1 µM). In the end, contractile response to KCl (80 mM) was tested, and arteries were bathed in a Ca^2+^-free PSS with sodium nitroprusside (10 µM) and papaverine (100 µM). Pressure steps from 10 to 125 mmHg were performed to determine the passive diameter of the arteries. Data from male (*n* = 7) and female (*n* = 7) were grouped since no significant difference was found between both genders.

### 4.8. Data Acquisition and Analyses

The recording of myography and arteriography data was performed with AcqKnowledge software (Biopac, Goleta, CA, USA). Statistical analysis and graphs were made with GraphPad Prism 7.02 software (La Jolla, CA, USA). The non-parametric Mann–Whitney test, Friedman, and two-way ANOVA (mixed effect model) analysis followed by Holm–Sidak’s multiple comparison tests were used when appropriated (ns: not significant, * for *p* < 0.05, ** for *p* < 0.01, *** for *p* < 0.001, **** for *p* < 0.0001). Data were shown as mean ± SEM. Since no significant differences by appropriate statistical tests (Mann–Whitney or multiple *t*-tests corrected using the Holm–Sidak method) were observed between females and males, the data from both genders (RT-qPCR, myography, and arteriography) were pooled.

## 5. Conclusions

Na_V_ channels are expressed in murine aorta and MA, in both endothelium and media. The Na_V_ channel genes *scn2a* and *scn1b* are the most abundant transcripts, indicating that murine arterial Na_V_ channels contain Na_V_1.2 channel with Na_V_β1. TTX strongly decreases contractile responses of MA, a model of resistance arteries, and increases their dilation responses to flow. This reveals the contribution of vascular Na_V_ channels to vascular tone regulation and could explain the decrease of total peripheral resistance during mammal tetrodotoxication.

## Figures and Tables

**Figure 1 marinedrugs-21-00196-f001:**
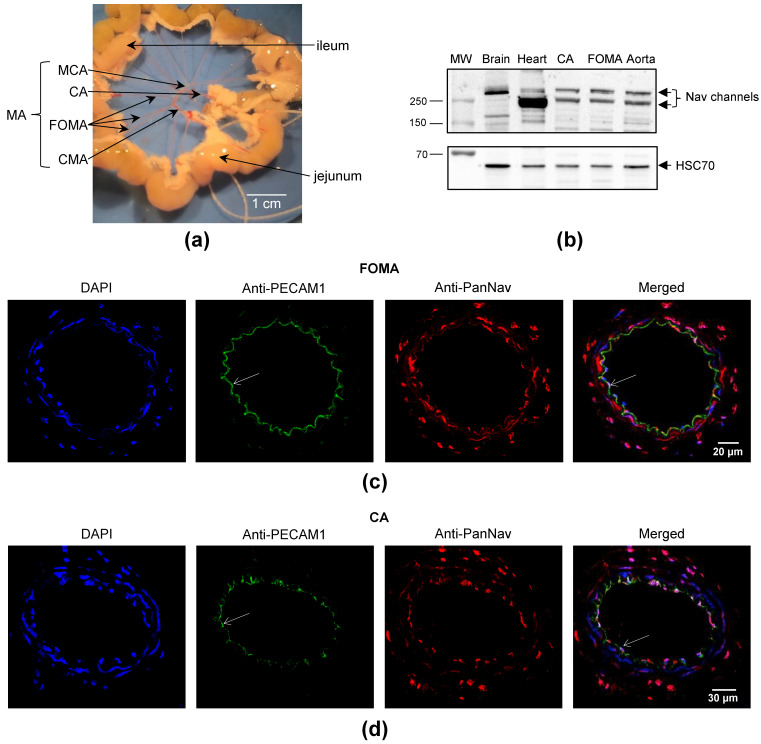
Expression of Na_V_ channels in mesenteric arteries. (**a**) The image illustrates the vascular bed of the small intestine in C57Bl/6J mouse. The small intestine (jejunum followed by ileum) was dissected and prepared to show MA. The two different branches of MA used in this work are indicated with arrows: CA and FOMA, cranial MA (CMA), and middle colic artery (MCA). (**b**) Western blot using an anti-PanNa_V_ antibody. HSC70 was used as a loading control and mouse brain and heart as positive controls. The experiments were carried out with 10 µg of proteins. (**c**,**d**) Representative immunofluorescent images are shown to illustrate the immunolabeling of intima and media in FOMA and CA. The nuclei were labeled with DAPI (in blue). The mesenteric vascular endothelium was immunolabeled with anti-PECAM1 antibodies (in green). Na_V_ channels were detected with an anti-PanNa_V_ antibody (PanNa_V_, in red). The overlay images (Merged) are shown on the right panels. The arrows indicate the endothelium.

**Figure 2 marinedrugs-21-00196-f002:**
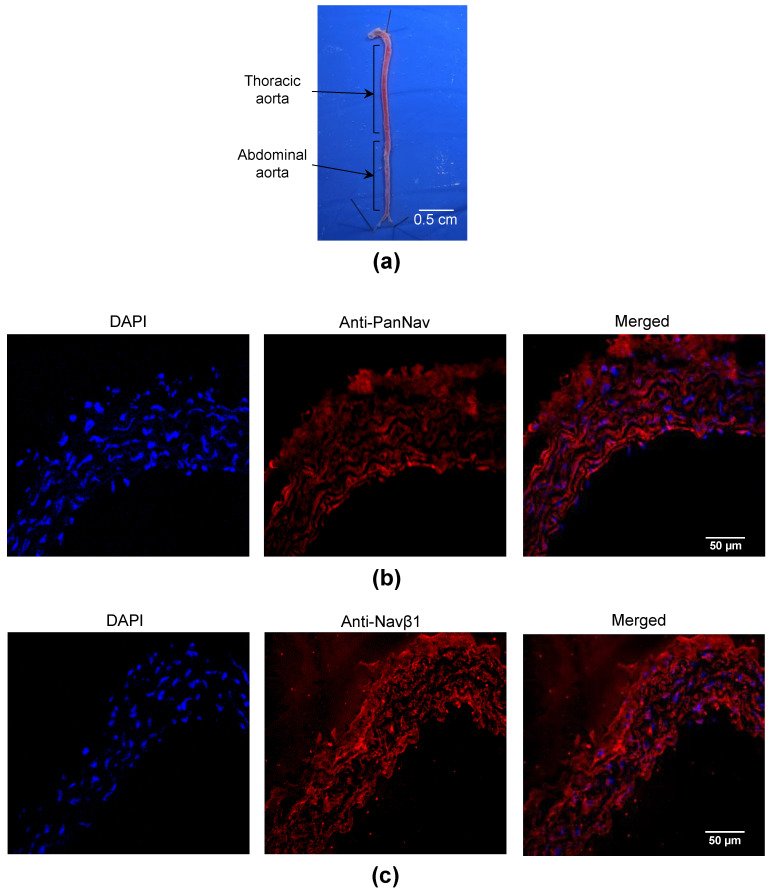
Immunolocalization of Na_V_ channels in mouse aorta. (**a**) The image illustrates an isolated aorta, which was dissected from a C57Bl/6J mouse. The thoracic aorta was used in this work. (**b**,**c**) Representative immunofluorescent images are shown to illustrate the expression of Na_V_ channel subunits in aortae. The nuclei were labeled with DAPI (left panels in blue). (**b**) The α-subunit was immunolocalized with an anti-PanNa_V_ antibody (in red). (**c**) The Na_V_β1 subunit was immunodetected with anti-Na_V_β1 antibodies (in red). The overlay of separated images (Merged) is shown on the right panels.

**Figure 3 marinedrugs-21-00196-f003:**
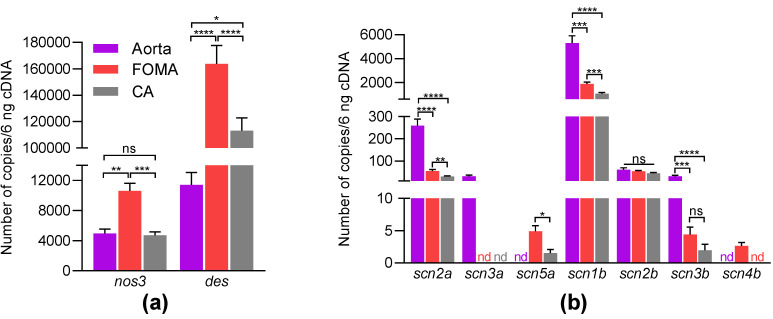
Gene expression profiles of Na_V_ channels in murine aorta and mesenteric arteries. The expression levels of *nos3*, *desmin* gene (*des*), (**a**) and Na_V_ channel genes (**b**) were determined by absolute RT-qPCR in aorta, FOMA, and CA. cDNA were prepared with RNA extracted from males (*n* = 5–6) and females (*n* = 6). Data are shown as mean ± SEM and were analyzed using two-way ANOVA corrected with Holm–Sidak’s test for multiple comparisons, ns: non-significant, * *p* < 0.05, ** *p* < 0.01, *** *p* < 0.001, **** *p* < 0.0001 (*p*-values are shown in Table 1). nd: not detected.

**Figure 4 marinedrugs-21-00196-f004:**
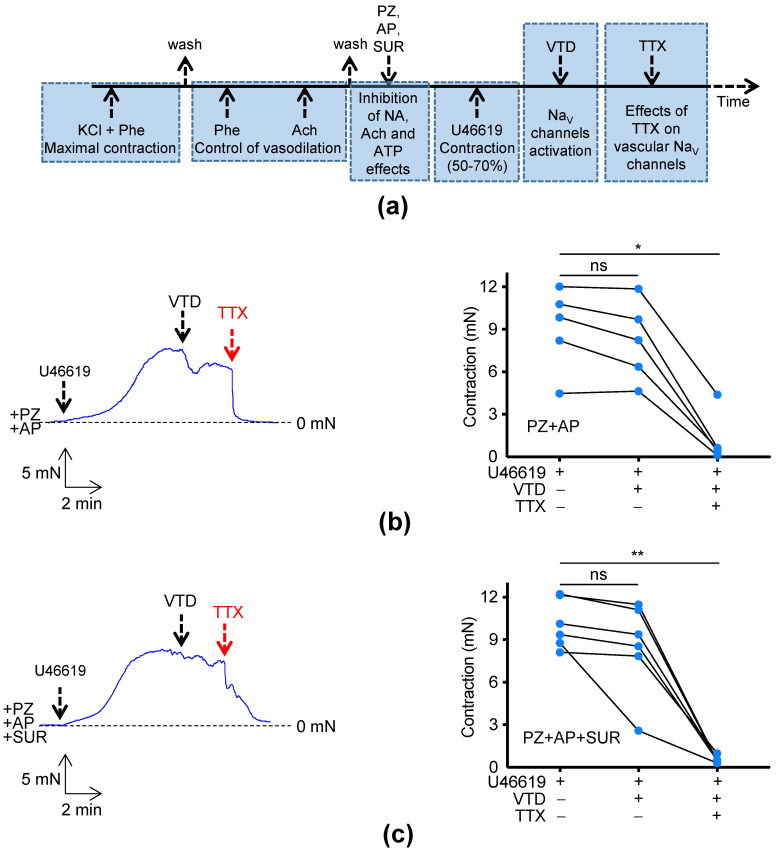
TTX induces vasorelaxation of mesenteric arteries. (**a**) The flow chart illustrates the experimental strategy used to unmask the effect of TTX on isolated CA of male C57Bl/6J mice. (**b**) The left panel shows a representative myograph trace, illustrating TTX-induced vasorelaxation, after treatment with PZ, AP, U46619, and VTD. (**c**) The left panel shows a representative myograph trace, illustrating TTX-induced vasorelaxation in the presence of SUR. As in (**b**), CA was treated with PZ, AP, U46619, and VTD. (**b**,**c**) The graphs on right represent connected scatter plot of individual values (contraction levels in mN) for each artery. Significance of differences between groups was evaluated with Friedman test (ns: not significant; * *p* < 0.05; ** *p* < 0.01).

**Figure 5 marinedrugs-21-00196-f005:**
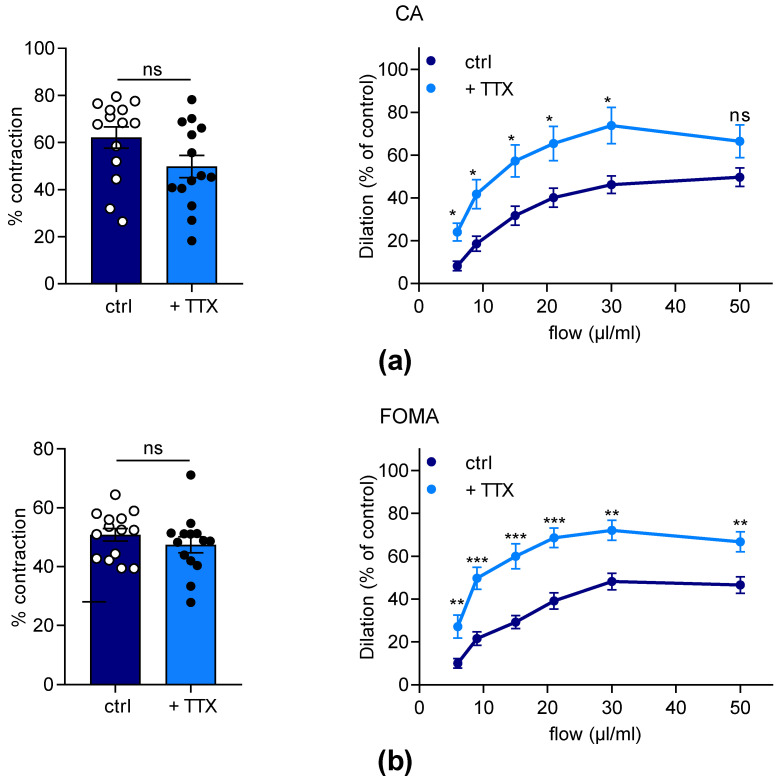
Effects of TTX on flow-mediated dilation of mesenteric arteries. Flow-mediated dilation was evaluated in both CA (**a**) and FOMA (**b**) from C57Bl/6J mice by arteriography in the absence (ctrl) or the presence of 1 µM TTX (+TTX). The left panels illustrate the scatter plots of the initial contraction level induced by Phe (1 µM) before flow rate increase. Mann–Whitney test was used to compare groups. The graphs showed in the right panels illustrate the flow–dilation relationship. Significance of difference between data was analyzed by mixed effect test followed by Holm–Sidak’s comparison test (α = 0.05). Data obtained from male (*n* = 7) and female (*n* = 7) were mixed. The values are means ± SEM (*n* = 14). ns: not significant; * *p* < 0.05, ** *p* < 0.01, *** *p* < 0.001.

**Table 1 marinedrugs-21-00196-t001:** Comparison of *nos3*, *desmin*, and Na_V_ channel gene expression between aorta, FOMA, and CA.

Gene Name	Protein Name	Arteries	Number of Genes (Mean ± SEM, *n* = 11–12)	Pairwise Comparison	*p*-Value
*scn2a*		aorta	258.58 ± 30.51	aorta vs. FOMA	<0.0001
	Na_V_1.2	FOMA	54.5 ± 6.9	aorta vs. CA	<0.0001
		CA	30.7 ± 2.2	FOMA vs. CA	0.0051
*scn3a*		aorta	30.9 ± 5.8		
	Na_V_1.3	FOMA	nd		
		CA	nd		
*scn5a*		aorta	nd		0.0005
	Na_V_1.5	FOMA	4.9 ± 0.9	FOMA vs. CA	0.022
		CA	1.5 ± 0.6		0.0122
*scn1b*		aorta	5300.9 ± 609.6	aorta vs. FOMA	0.0003
	β1-subunit	FOMA	1879.4 ± 157.7	aorta vs. CA	<0.0001
		CA	1069.5 ± 93.2	FOMA vs. CA	0.0008
*scn2b*		aorta	60.8 ± 9.2	aorta vs. FOMA	0.5517
	β2-subunit	FOMA	54.2 ± 3.2	aorta vs. CA	0.2275
		CA	45.3 ± 3.1	FOMA vs. CA	0.1176
*scn3b*		aorta	31.8 ± 4.1	aorta vs. FOMA	0.0001
	β3-subunit	FOMA	4.4 ± 1.2	aorta vs. CA	<0.0001
		CA	2.0 ± 0.9	FOMA vs. CA	0.1134
*Scn4b*		aorta	nd		
	β4-subunit	FOMA	2.7 ± 1.2		
		CA	nd		
*nos3*		aorta	4961.4 ± 585.6	aorta vs. FOMA	0.0017
	eNOS	FOMA	10,619.8 ± 1007.8	aorta vs. CA	0.7544
		CA	4721.6 ± 449.2	FOMA vs. CA	0.0008
*des*		aorta	11,414.8 ± 1644.2	aorta vs. FOMA	<0.0001
	desmin	FOMA	163,847.8 ± 13,848.2	Aorta vs. CA	<0.0001
		CA	113,193.8 ± 9609.1	FOMA vs. CA	0.0156

The gene copy numbers were evaluated for 6 ng cDNA. nd: not detected.

**Table 2 marinedrugs-21-00196-t002:** Sequence of primers used for RT-qPCR.

Gene Name	GenBank Accession Number	Forward Primer (5′−3′)	Reverse Primer (5′−3′)
*scn1a*	NM_018733.2	ttgcaaggggcttctgttta	aggtccacaaactccgtcac
*scn2a*	NM_001099298.2	gggttgcatatggtttccaa	cccaaggcatttgcagtta
*scn3a*	NM_018732.3	tcctcagtagtggtgctttgg	gatgtaagtgaagactttgtcagca
*scn4a*	NM_133199.2	gaaaaccatcacggtcatcc	tccgagagctttttcacagac
*scn5a*	NM_021544.4	gccagatctctatggcaacc	ttgcccttattcagcacgat
*scn8a*	NM_001077499.2	ctggtgctggttggacttc	gcccagggcattagctataa
*scn9a*	NM_001290674.1	gctgagcctatcaatgcaga	acttggcagcatggaaatct
*scn10a*	NM_001205321.1	tgggtagcttatggcttcaaa	ctatgaggcttgtgagggaga
*scn11a*	NM_011887.3	ttcataatgtgtggcaactgg	ttattgcacgtggaaccatc
*nos3*	NM_008713.4	ccagtgccctgcttcatc	gcagggcaagttaggatcag
*pecam1*	NM_008816.3	ccagtgccctgcttcatc	gcagggcaagttaggatcag

## Data Availability

Not applicable.

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
