# Peer review of "Tetrodotoxin Decreases the Contractility of Mesenteric Arteries, Revealing the Contribution of Voltage-Gated Na+ Channels in Vascular Tone Regulation"

_marinedrugs, 2023, doi:10.3390/md21030196_

Round 1

Reviewer 1 Report

Comments to the Authors

 The manuscript entitled “Tetrodotoxin decreases the contractility of mesenteric arteries, revealing the contribution of voltage-gated Na+ channels in vascular tone regulation“ (Manuscript ID: marinedrugs-2266339) describes the presence of various subunits of voltage-gated sodium channel proteins and mRNA in endothelium and smooth muscle cells of mouse aorta and mesenteric arteries. Moreover, it shows that tetrodotoxin induce vasorelaxation and potentiate flow-mediated dilation response acting directly on the Nav channels of the vascular wall, which might add a further mechanism to the development of severe hypotension observed in tetrodotoxications.

The study is interesting, well-designed with high quality manuscript, containing only few misspellings and overall easy to follow except for a part mentioned later. The graphs are good quality, some minor suggestions will be mentioned later. The number of experiments is sufficient and the analysis of results is satisfactory. The conclusions are mostly supported by the results, which are compared with the literature by the 36 references.

 Major comments/questions/suggestions:

1.      The description of results with transcript levels is really hard to understand. First of all in Table 1 I do not know what the mean values represent, how can some of those be negative? Also on figure 3 the letter code of significance is not the best way to indicate the statistical differences. It would be much easier to follow if you simply indicate which column is significant compared to which other(s).

2.      Did you try testing the actions of TTX on endothel denuded vessels to find out if it targets endothelium and/or vascular smooth muscle? Related to this I do not understand the explanation at the end of page 9 where the vasorelaxant effects of TTX observed in MA in response to flow increase are discussed with endothelial Nav channels as a possible target of the TTX. According to my knowledge the sheer stress from the flow increase would lead to activation of endothelial Nav channels which depolarize those cells thereby leading to increase of intracellular Ca2+ concentration. That would activate NO synthesis leading to relaxation of the adjacent VSMC-s causing vasodilation. Base on that if one assumes that TTX reduces the activity of endothelial Nav channels the vasodilation response to flow increase should be reduced and not increased as shown in the results. Please clarify.

3.      Did you try to contract the vessels with another substance apart from U46619 to test the effects of TTX in the presence of VTD? Why did you choose the stimulation of thromboxane A2 mediated pathway? Moreover, at the end of chapter 4.6 it is described that TTX was applied in the presence of VTD and then it tells that this was repeated in the presence of U46619. So clarify please if TTX effect was indeed tested in the presence of VTD without causing contraction with U46619 or anything else?

4.      Authors mention: “contribution of Nav channels in contractile response might occur in non physiological or pathological conditions,”. In what conditions do you have in mind and how is that condition related to tetrodotoxications. Is the tetrodotoxication induced hypotension present in healthy individuals? Please add that to the discussion.

5.      Did you perform western blot from aorta samples? It would be nice to compare those results with PCR results (as done for MA in lines 215 and 216).

6.      How can one be sure that Nav channels are present in intima layer of aorta without Anti-PECAM1 staining?

7.      Did you try to remove the vasorelaxant action of TTX by washout? Was the action of TTX reversible?

8.      Did you test gender difference of the observed findings before pooling the results from male and female animals together?

9.      Why did you use suramin only in some experiments and just prazosin and atropine in others?

 Minor comments/questions/suggestions:

1.      In line 18 the following: “…symptoms, among which a severe hypotension.” sounds strange so I suggest to change that to: “…symptoms including severe hypotension.”

2.      In line 26 “…indicating…” could be changed to “…suggesting…” as there is no necessary causative relationship.

3.      In line 42 what do you mean by “electrical-excitability coupling”? Are those two not the same events? I only heard about excitation-contraction coupling or pharmaco-mechanical coupling…

4.      On figure 1a “ileon” should be corrected to “ileum” according to English terminology.

5.      On page 5 please unify stating the number of gene copies for desmin, nos3 and Nav subunit transcripts. For the former two there is a range but for the Nav channels it is given as average+/-SEM.

6.      On figure 4 please indicate the 0 mN level on the representative graphs and remove the “(n=5)” and “(n=6)” from those on panel b and c, respectively.

7.      In line 176 please mention the meaning of FMD at its first appearance.

8.      The sentence in lines 199-200 should be rephrased as the specific antagonists (I guess it refers to prazosin, atropine +/-suramin) will not prevent the neurotransmitter release, only the effects of the released transmitters on the vessel diameter.

9.      The doi numbers for the following references should be added: 1., 17., 22., and 34.

10.  Why the second half of the title of reference 14 is written with CAPITAL letters?

11.  Please check the end of reference 34.

12.  In reference 36 “et al.” could be replaced by the name of the last author, as only one name is not mentioned.

Misspellings:

1.      In line 50 “…endothelial that…” should be “…that endothelial…”.

2.      In line 52 an “e” is missing from the end of “interfer”.

3.      In line 53 “…coupled…” should be “…coupling…”.

4.      In line 61 an “d” is missing from the end of “overcome”.

5.      In line 309 “4.4. Western blot” should be “4.3. Western blot”.

6.      In line 375 “were” should be “was”.

7.   In lines 429-431 I guess there is no need for the text in the Acknowledgments part.

 Based on my previous comments I recommend at least a thorough minor revision of the manuscript.

Author Response

Dear Reviewer 1,

thank you very much for your work. You help us to improve our manuscript. We addressed almost all your questions and points as indicated below.

Sincerely yours

Professor Christian Legros

Reviewer 1

Major comments/questions/suggestions:

  1. The description of results with transcript levels is really hard to understand. First of all in Table 1 I do not know what the mean values represent, how can some of those be negative? Also on figure 3 the letter code of significance is not the best way to indicate the statistical differences. It would be much easier to follow if you simply indicate which column is significant compared to which other(s).

Thanks for your comments. We simplified the paragraph that describes our RT-qPCR data.

There were no negative values. In this column, we reported the difference between pairwise comparison. However, we apologise for multiple typing errors. Thus, we build a new table 1 in which we reported all expression levels (mean±SEM). On figure 3, we changed the way to show the statistical differences as you proposed. Consecutively, the corresponding caption was modified.

  1. Did you try testing the actions of TTX on endothel denuded vessels to find out if it targets endothelium and/or vascular smooth muscle? Related to this I do not understand the explanation at the end of page 9 where the vasorelaxant effects of TTX observed in MA in response to flow increase are discussed with endothelial Nav channels as a possible target of the TTX. According to my knowledge the sheer stress from the flow increase would lead to activation of endothelial Nav channels which depolarize those cells thereby leading to increase of intracellular Ca2+ concentration. That would activate NO synthesis leading to relaxation of the adjacent VSMC-s causing vasodilation. Base on that if one assumes that TTX reduces the activity of endothelial Nav channels the vasodilation response to flow increase should be reduced and not increased as shown in the results. Please clarify.

Your comments are important, but we think we have discussed this point in lanes 260-267 (first submission).

In myography experiments, it would be very interesting to assay the effects of TTX without endothelium or in the presence of an inhibitor of eNOS, or of the EDHF pathway. Instead, we carried out arteriography experiments, in which we explored the effects of TTX on endothelium function. In these experiments, the myogenic tone is constant, while the increasing flow only activates endothelium response. To clarify this, we changed the first sentence of the paragraph related to arteriography in the results section as followed:

“In order to investigate the effects of TTX on endothelial-mediated dilation, we performed arteriography experiments on isolated FOMA and CA (Figure 5).”

There is only one study that shows that under shear-stress, the endothelial Nav channels are activated, allowing the reduction of MapKinase activation (Traub et al., doi:10.1074/jbc.274.29.20144, ref 16). However, this has been shown with HUVEC in in vitro experiments. Subesequent increase of intracellular Ca2+ concentration, after Nav channel activation has not been proven in the litterature in endothelial cells. If I have missed an article, please provide me the reference, I will take this into account in the discussion. In an another study that we have in revision in IJMS, we applied veratridine on EC loaded with FURA-2, but we did not observed any intracellular Ca2+ increase. Since in arteriography, we found that TTX induced a vasodilation or potentiated the flow-mediated response, we think that the contribution of endothelial Nav channels is not obvious, but we could not exclude it. In arterial myocytes, the crosstalk between Nav channels and Ca2+ signalling has been clearly established (Ho et al., 2013; Boccara et al, 1999). Thus, the activation of Nav channels in myocytes leads to contraction, which will be abolish by TTX. Concerning endothelial Nav channels, the mechanism is more complicated. Our data from arteriography, clearly suggest that they contribute to shear-stress response. Based on the data showed by Traub et al. (1999), the shear-stress activates Nav channels that downregulate erk1/2 phosphorylation. Since erk1/2 favors eNOS activation, this could explain why TTX increases vasodilation. However, the regulation of eNOS by erk1/2 is controversial.

Accordingly, we changed the corresponding paragraph in the discussion.

  1. Did you try to contract the vessels with another substance apart from U46619 to test the effects of TTX in the presence of VTD? Why did you choose the stimulation of thromboxane A2 mediated pathway? Moreover, at the end of chapter 4.6 it is described that TTX was applied in the presence of VTD and then it tells that this was repeated in the presence of U46619. So clarify please if TTX effect was indeed tested in the presence of VTD without causing contraction with U46619 or anything else?

In myography experiments, we wanted to contract the arteries with a mediator that gives repoducible results and also that could be used in the presence of antagonists of alpha1-adrenoreceptors. In the lab, we have done a lot of experiments with U46619 that activates the intracellular Ca2+ signaling. U46619 has also extensively used for these kind of experiments. We have We have also planned to use angiotensine II or serotonine. Since this mediator also activates similar intracellular pathway, it could led to similar results. KCl contraction was excluded because it depolarizes both smooth muscle cells and terminal nerve endings. In 4.6 subsection, there is an error. The protocol is shown in figure 4a. The corrected protocol in 4.6 is:

“Two cocktails of antagonists were then added in order to prevent any effects of neurotransmitter release (NA, Ach and ATP) by terminal nerve endings: 1 µM PZ with 100 nM AP (PZ+AP) or 1 µM PZ with 100 nM AP with 100 µM SUR (PZ+AP+SUR). Then, VTD (30 µM) was applied to activate all NaV channels in arterial tissue. After stabilization, TTX (1 µM) were applied. Afterwards, U46619 (10 to 80 nM, thromboxane A2 analogue) was applied to reach 50 to 70% of maximal contraction, followed by VTD at 30 µM. After the stabilization of VTD-induced effects, 1 µM TTX was added in the bath.”

Thus, this section was modified accordingly. Sorry for this mistake and thanks for your comments.

  1. Authors mention: “contribution of Nav channels in contractile response might occur in non physiological or pathological conditions,”. In what conditions do you have in mind and how is that condition related to tetrodotoxications. Is the tetrodotoxication induced hypotension present in healthy individuals? Please add that to the discussion.

We believe as Ho et al (2013) and Virsolvy et al (2021), that in normal or physiological conditions, Nav channels are inactive or silent and do not contribute to physiological vascular responses. However, since they could induce intracellular Ca2+ responses in myocytes through coupling with NCX or other Ca2+ transporters, they could participate to the vascular tone in abnormal conditions. Interestingly, Virsolvy et al (2021) have shown that hypoxic conditions allow the activation of silent vascular Nav channels, leading to spontaneous contractions (vasomotion). Since respiratory issues (dyspnea, hypoxia) occur during tetrodotoxication, this could favor the activation of these silent vascular Nav channels, however the presence of TTX will block this response to hypoxia. We removed the following two sentences “However, this vasorelaxation effect appeared when arteries were strongly contracted with U46619 and treated with VTD. Thus, we believe that the contribution of NaV channels in contractile response might occur in non physiological or pathological conditions, as it has been demontrated in human uterine arteries under hypoxia [15].” And we changed the last paragraph of the discussion to address your point.

  • Did you perform western blot from aorta samples? It would be nice to compare those results with PCR results (as done for MA in lines 215 and 216).

Indeed, our western blot also has a lane with protein extract from aorta. We modified Figure 1b to show it. However, we also observed two bands as in FOMA and CA. We have probably misinterpretated the lighter band (250 kDa), since in fact, its MW is a bit higher than observed in heart. We changed the corresponding paragraph. We thought that this band could correspond to the isoform X2 (GenBank: XP_006498646.1). The size difference between this isoform X2 and the variant a (GenBank: AIW80039.1) is 28.4 kDa without glycan chains, which matches with the difference between both bands immunolabeled on our blot.

  1. How can one be sure that Nav channels are present in intima layer of aorta without Anti-PECAM1 staining?

We re-examined our histoimmunochemistry data. It seems that there is no clear evidence that anti-PanNav immunolabels intima. The use of anti-PECAM1 staining would be required to confirm this. We changed the text in the results and discussion section.

  1. Did you try to remove the vasorelaxant action of TTX by washout? Was the action of TTX reversible? Yes, the vasorelaxation effect induced by TTX was fully reversible. After TTX application and subsequent washout, we could perform the myography protocole (figure 4.a) once again and we obtained similar responses. We added the following sentence to state this point in corresponding paragraph : “In both cases, after washout, CA gave similar responses to U46619 and TTX”.
  2. Did you test gender difference of the observed findings before pooling the results from male and female animals together?

Concerning myography and arteriography, we first did a Mann and Whitney to compare artery diameters in the presence of phenylephrine between male and female in control condition and with TTX. There were no significant differences between both genders. Then, we compared artery diameters at each flow between males and females in control condition and with TTX (multiple t tests corrected using the Holm-Sidak method). No significant difference was observed between group. We added a sentence in the text to precise this (material and method section 4.8).

  1. Why did you use suramin only in some experiments and just prazosin and atropine in others?

We used suramin in order to eventually eliminate the effect of ATP release by terminal nerve endings that can induce vasorelaxation. There was non significant differences without or with ATP. We added a sentence to precise this point in section 2.1.

Minor comments/questions/suggestions:

  1. In line 18 the following: “…symptoms, among which a severe hypotension.” sounds strange so I suggest to change that to: “…symptoms including severe hypotension.” Corrected.
  2. In line 26 “…indicating…” could be changed to “…suggesting…” as there is no necessary causative relationship. Corrected.
  3. In line 42 what do you mean by “electrical-excitability coupling”? Are those two not the same events? I only heard about excitation-contraction coupling or pharmaco-mechanical coupling… Corrected.
  4. On figure 1a “ileon” should be corrected to “ileum” according to English terminology. Corrected.
  5. On page 5 please unify stating the number of gene copies for desmin, nos3 and Nav subunit transcripts. For the former two there is a range but for the Nav channels it is given as average+/-SEM. Values are now in Table 1.
  6. On figure 4 please indicate the 0 mN level on the representative graphs and remove the “(n=5)” and “(n=6)” from those on panel b and c, respectively. The figure 4 was modified.
  7. In line 176 please mention the meaning of FMD at its first appearance. Corrected.
  8. The sentence in lines 199-200 should be rephrased as the specific antagonists (I guess it refers to prazosin, atropine +/-suramin) will not prevent the neurotransmitter release, only the effects of the released transmitters on the vessel diameter. Corrected.
  9. The doi numbers for the following references should be added: 1., 17., 22., and 34. Corrected.
  10. Why the second half of the title of reference 14 is written with CAPITAL letters? Corrected.
  11. Please check the end of reference 34. I checked it. It seems correct.
  12. In reference 36 “et al.” could be replaced by the name of the last author, as only one name is not mentioned. Corrected.

Misspellings:

  1. In line 50 “…endothelial that…” should be “…that endothelial…”. corrected
  2. In line 52 an “e” is missing from the end of “interfer”. corrected
  3. In line 53 “…coupled…” should be “…coupling…”. corrected
  4. In line 61 an “d” is missing from the end of “overcome”. We checked the grammar and since overcome is irregular (overcome-overcame), it should be “overcome”.
  5. In line 309 “4.4. Western blot” should be “4.3. Western blot”. corrected
  6. In line 375 “were” should be “was”. corrected
  7.  In lines 429-431 I guess there is no need for the text in the Acknowledgments part.

We filled this section.

Reviewer 2 Report

The manuscript submitted to Marine Drugs by Park et al., succinctly shows an interesting role for TTX-sensitive VGSCs in vasculature tone regulation; a process the authors speculate contributed to TTX-toxicity. The manuscript data, experimental flow, and interpretation is appropriate.  The data nicely show the distribution of specific VGSCs, which is bolstered by pan-antibody staining, as well as an interesting pharmacological paradigm to expose VGSC contribution to vascular tone. The manuscript is polished and a pleasure to read. The comments are listed:

1) Please address the western profile in respect to RNA profiles. For example, the proteins bands for what is hypothesized to be Nav1.5 in heart and in CA/FOMA doesn’t approximate the relationship expected following RNA quantification using RT-qPCR.

2)  The immunohistochemical VGSC profile in the FOMA and CA appears quite strong beyond the vascular endothelium PECAM1 labeled cells. Please address this finding.

3) Would it be useful to show that TTX added before the VTD blocks the functional change in contractility shown in Fig 4?

Minor comments:

1) A simple sentence clearly stating why the selected TTX sensitive VGSC isoforms are probed would be useful for naïve readers.

2) Any logical reason why the data in Table 1 doesn’t match Fig 3? If so, please elaborate in a short sentence.

3) Were the male and female mice pooled for tissue extraction because previous work established limited sexual differences in VGSC expression distribution (in other tissues)?

4) Additional information explaining statistical comparisons in Fig 3 would be helpful. For example, how could the letter “a” indicate no statistical significance based on AVG SEM shown? I think I must be miss-reading this in the legend! Please add a sentence stating which data is not shown since it was undetectable directly in Fig 3 legend for those that focus on figures and not Results text.

5) Protocol in Fig 4C could benefit from additional explanation in legend.

6) Could the authors please comment on how VGSC contributes to the observed functional regulation of vasculature tone in regards to membrane potential, as well as the contribution of VGSC voltage-dependent inactivation?  

7) Consider changing word intoxication LINE 204 to TTX poisoning or the other term you use tetrodotoxification.

8) Would it be worth briefly expounding on VGCS isoform differences in mouse versus human in Discussion (voltage-gating properties etc)?

9) Might be useful to comment in Methods why a [TTX] in microM range for the TTX-S VGSCs was used (when you state in intro TTX sensitivity of TTX-R is micromolar).

Author Response

Dear Reviewer 2,

we would like to thank you for constructive reviewing and you will find below the responses in red to your questions. 

Sincerely yours,

Professor Christian Legros

  • Please address the western profile in respect to RNA profiles. For example, the proteins bands for what is hypothesized to be Nav1.5 in heart and in CA/FOMA doesn’t approximate the relationship expected following RNA quantification using RT-qPCR.

As we said to Reviewer 1, we have misinterpretated the western blot. The lighter band (250 kDa), is a bit higher than observed in heart. We changed the corresponding paragraph in the result section and also the discussion, according our RT-qPCR data. We thought that this band could correspond to the isoform X2 (GenBank: XP_006498646.1). The size difference between this isoform X2 and the variant a (GenBank: AIW80039.1) is 28.4 kDa without glycan chains, which matches with the difference between both bands immunolabeled on our blot.

2)  The immunohistochemical VGSC profile in the FOMA and CA appears quite strong beyond the vascular endothelium PECAM1 labeled cells. Please address this finding.

The cells in red beyond cells labelled by anti-PECAM1 antibody could be due to the expression of Nav channels in myocytes (media). We added a sentence at the end of the firts paragraph of section 2.1.

  • Would it be useful to show that TTX added before the VTD blocks the functional change in contractility shown in Fig 4?

We did not perform this control. Since both ligands are specific for Nav channels, we believe that at 1 µM, TTX will render all Nav channels silent, and thus VTD, either at 30 µM, could not activate them.

Minor comments:

  • A simple sentence clearly stating why the selected TTX sensitive VGSC isoforms are probed would be useful for naïve readers.

We agree with this comment. Thus, we inserted the Nav channel subtypes which are TTX-sensitive and TTX-resistant in the 3rd paragraph of the introduction. We added a sentence to precise the putative composition of Nav channel expressed in aorta and mesenteric arteries, deduced from RT-qPCR data (last sentence of 3rd paragraph of the section 2.1).

2) Any logical reason why the data in Table 1 doesn’t match Fig 3? If so, please elaborate in a short sentence.

This Table 1 was corrected.

  • Were the male and female mice pooled for tissue extraction because previous work established limited sexual differences in VGSC expression distribution (in other tissues)?

RNA and then cDNA from both genders were used separatly. Before being pool, a Mann and Whitney test had been run to analyse the difference between qPCR data from males and females. We added a sentence to precize this point in the material and method section (4.8).

  • Additional information explaining statistical comparisons in Fig 3 would be helpful. For example, how could the letter “a” indicate no statistical significance based on AVG SEM shown? I think I must be miss-reading this in the legend! Please add a sentence stating which data is not shown since it was undetectable directly in Fig 3 legend for those that focus on figures and not Results text.

As asked by Reviewer 1; statistical data comparisons were changed in Fig. 3. Consecutively, the corresponding caption was modified.

5) Protocol in Fig 4C could benefit from additional explanation in legend. As asked, we modified the caption of Fig. 4.

6) Could the authors please comment on how VGSC contributes to the observed functional regulation of vasculature tone in regards to membrane potential, as well as the contribution of VGSC voltage-dependent inactivation?  

We discussed this point in the last paragraph of the discussion.

7) Consider changing word intoxication LINE 204 to TTX poisoning or the other term you use tetrodotoxification. We changed “intoxication” by “tetrodotoxification”.

8) Would it be worth briefly expounding on VGCS isoform differences in mouse versus human in Discussion (voltage-gating properties etc)? This is an interesting point, we thought it would be very speculative. The methods used for the detection of Nav channel gene transcripts are not the same than we used for murine arteries (absolute RT-qPCR) and in human uterine arteries or aorta (classical RT-qPCR or relative RT-qPCR). In addition, the beta subunits were not always investigated. Moreover, it lacks information from same kind of arteries for discussing further.

9) Might be useful to comment in Methods why a [TTX] in microM range for the TTX-S VGSCs was used (when you state in intro TTX sensitivity of TTX-R is micromolar).

We agree with this point. Our RT-qPCR data suggested a low expression of Nav1.5, a TTX-R Nav channel with an IC50 is close to 1 µM (Durán-Riveroll and Cembella, 2017; Tsukamoto et al., DOI:10.1111/bph.13985). Thus, we expected that with 1 µM TTX we would have a partial effects on isolated arteries if this Nav1.5 is involved in vascular tone. On the other hand, at 1 µM TTX we were sure to totally inhibit arterial responses mediated by TTX-S Nav channels. We added the following sentence in results section to justify this choice : “Since both TTX-S and TTX-R NaV channel gene transcripts have been detected by RT-qPCR in MA, we chose 1 µM TTX in order to affect both kind of NaV channels. At this concentration, we expected partial effects if TTX-R NaV channels would contribute to arterial response or a full response mediated by TTX-S NaV channels.”